# Analysis of Selected Small Proline-Rich Proteins in Tissue Homogenates from Samples of Head and Neck Squamous Cell Carcinoma

**DOI:** 10.3390/diagnostics15131633

**Published:** 2025-06-26

**Authors:** Dariusz Nałęcz, Agata Świętek, Dorota Hudy, Zofia Złotopolska, Jakub Opyrchał, David Aebisher, Joanna Katarzyna Strzelczyk

**Affiliations:** 1Department of Otolaryngology and Maxillofacial Surgery, St. Vincent De Paul Hospital, 1 Wójta Radtkego St., 81-348 Gdynia, Poland; zzlotopolska@szpitalepomorskie.eu; 2Department of Medical and Molecular Biology, Faculty of Medical Sciences in Zabrze, Medical University of Silesia in Katowice, 19 Jordana St., 41-808 Zabrze, Poland; agata.swietek@sum.edu.pl (A.Ś.); dorota.hudy@sum.edu.pl (D.H.); jstrzelczyk@sum.edu.pl (J.K.S.); 3Silesia LabMed Research and Implementation Centre, Medical University of Silesia in Katowice, 19 Jordana St., 41-808 Zabrze, Poland; 41st Department of Oncologic Surgery, Maria Skłodowska-Curie National Research Institute of Oncology, 15 Wybrzeże Armii Krajowej St., 44-102 Gliwice, Poland; 5Department of Photomedicine and Physical Chemistry, Medical College of the University of Rzeszów, 35-310 Rzeszów, Poland; daebisher@ur.edu.pl

**Keywords:** head and neck squamous cell carcinoma, HNSCC, carcinogenesis, protein level, tumour, surgical margin, small proline rich proteins, SPRRs, SPRR1A, SPRR2A

## Abstract

**Background/Objectives**: Head and neck squamous cell carcinoma (HNSCC) ranks sixth in the world in terms of incidence. Small proline-rich proteins (SPRRs) are precursors of the keratinocyte envelope and act as substrates of transglutaminase. A change in SPRR expression is characteristic in a few types of cancer. Our aim was to determine the concentration of SPRR1A and SPRR2A in tumours samples obtained from 61 patients with HNSCC (OSCC, OPSCC, LSCC, HPSCC, NCSCC, and SSCC). Also, we aimed to determine the relationship between protein concentration and other clinical and/or demographic variables. **Methods:** An ELISA test was used to determine the concentrations of SPRR in the tumour tissue homogenates. **Results**: In margin samples, we found a statistically significant association between SPRR1A levels and nodal status (N) and between SPRR1A levels in tumours and margins with G2 histological grade. When we analysed the effect of tobacco and alcohol habits, we found a statistically significant difference between the SPRR1A and SPRR2A amount in smokers and non-smokers in margin samples. Also, we found a statistically significant difference between the SPRR1A and SPRR2A levels in tumour and margin samples obtained from patients that either abstain and occasionally or regularly consume alcohol. Furthermore, we found in tumour and margin samples from patients with concomitant diseases an association between SPRR1A and SPRR2A levels. Our results showed altered concentrations of SPRR1A at margins, depending on HPV status. **Conclusions**: These results suggest that differences in SPRR proteins are determined by disease status and unhealthy behaviours, which, in a wider perspective, can influence carcinogenesis.

## 1. Introduction

Head and neck squamous cell carcinoma (HNSCC) occurs in 95% of all head and neck types of cancer. Additionally, HNSCC is the 6th most prevalent cancer overall [1]. HNSCC is known as a group of cancers that occur in various anatomical structures, including the oral cavity, pharynx, and larynx [2]. HNSCCs are an increasingly serious clinical problem. Each year, more than 600,000 new cases are reported worldwide, with a mortality rate between 40 and 50% [3]. Exogenous factors associated with the risk of HNSCC include exposure to tobacco smoke, alcohol abuse, poor diet, and poor oral hygiene, as well as an infection by human papillomavirus (HPV). Other potential risk factors can be designated as genetic factors in origin [1,4].

Small Proline-Rich Protein 1A (SPRR1A) and 2A (SPRR2A) are members of the Small Proline-Rich Protein (SPRR) family of keratinocyte envelope precursor proteins. Lee et al. describe these molecules as specific markers of the differentiation of squamous epithelial cells and keratinocytes that function as transglutaminase substrates [5]. These proteins are also responsible for the formation of the skin barrier, and mutations in their genes can make the skin more susceptible to damage from harmful agents [6]. Recent reports have shown that SPRRs have even more versatile functions and are also involved in cell migration, induction of epithelial–mesenchymal transition, free radical quenching, prevention of chromosome damage, and p53 signalling. Furthermore, studies have shown that SPRR1A and SPRR2A proteins can exhibit antimicrobial activity by forming a protective skin barrier [7]. Decreased or elevated expression of SPRR1A and SPRR2A with prognostic significance has been observed in lymphoma, lung cancer, breast cancer, oesophageal cancer, laryngeal cancer, and oral cavity cancer [8,9,10,11,12]. To date, the roles of SPRR1A and SPRR2A in HNSCC have not been well understood. Measuring SPRR1A and SPRR2A levels is an important step towards understanding the molecular pathways of SPRRs. The levels of SPRR1A and SPRR2A proteins in HNSCC appear to be valuable due to their direct relationship with cellular differentiation processes and epithelial barrier maintenance. This represents an opportunity to identify new therapeutic targets and offers the possibility to use them as prognostic and predictive biomarkers, which may improve the personalisation of therapy and provide more effective treatment strategies.

It is hypothesised that the levels of SPRR1A and SPRR2A will be altered in the tumour and margin samples from patients with HNSCC, which will reflect their role in the processes of cancer cell proliferation and cancer cell survival. We speculate that the levels of SPRR1A and SPRR2A proteins in tumour samples will be significantly correlated with the following factors, such as selected demographic factors, smoking habit, alcohol consumption, and p16/HPV status.

In this study, we evaluated both levels of SPRR1A and SPRR2A proteins and matched the surgical margin in samples from patients with primary HNSCC. We also analysed the association of SPRR1A and SPRR2A levels with demographic factors, smoking status, alcohol consumption, p16, and HPV status.

## 2. Materials and Methods

### 2.1. Study Population

The study group included 61 (42 men, 19 women) patients with HNSCC. The clinical study was performed in the Department of Otolaryngology and Maxillary Surgery at St. Vincent De Paul Hospital in Gdynia, Poland. The main inclusion criteria were patients over 18 years of age, without preoperative radiotherapy or chemotherapy, with primary HNSCC, who consented to participate in the study. During hospitalisation, all patients underwent surgical resections. Tumour specimens, and surgical margin specimens at a distance of at least 10 mm from the tumour, were collected. The tumour samples were verified histopathologically as HNSCC, and the surgical margin samples were confirmed by a pathologist to be cancer free. The samples were transported on ice to the laboratory of the Department of Medical and Molecular Biology in Zabrze of the Medical University of Silesia. The study was approved by the Bioethical Committee of the Regional Medical Chamber in Gdansk (no. KB-42/21).

To prepare samples for evaluation of the SPRR1A and SPRR2A protein concentration, tumour tissue and surgical tissue margin samples were homogenised using a PRO Bio-Gen PRO200 homogeniser (PRO Scientific Inc., Oxford, CT, USA) at 10,000 rpm in nine volumes of PBS buffer (EURx, Gdansk, Poland). The homogenates were sonicated with a UP100H ultrasonifier (Hielscher, Teltow, Germany). The total protein level was determined using an AccuOrangeTM protein quantitation kit (Biotium, Fremont, CA, USA). Fluorescence was evaluated with a Synergy H1 microplate reader (BioTek, Winooski, VT, USA). To measure the levels of the proteins studied, an enzyme-linked immunosorbent assay (ELISA) was used, according to the manufacturer’s instructions. The SPRR1A concentration was evaluated using a human ELISA kit for Small Proline-Rich Protein 1A, Product No. SED802Hu (Cloud-Clone Corp., Houston, TX, USA), with a sensitivity of 0.063 ng/mL. The SPRR2A level was assayed using the human ELISA kit for Small Proline-Rich Protein 2A, Product No. SED804Hu (Cloud-Clone Corp., Houston, TX, USA), with a sensitivity of 0.057 ng/mL. The absorbances of the samples were measured using a Synergy H1 microplate reader (Bio-Tek, Winooski, VT, USA). The measurement was performed at a wavelength of 450 nm. The results obtained are reported as the total protein level in units of ng/µg of protein.

### 2.2. DNA Isolation, HPV, p16+, and Proliferative Index Ki-67 Confirmation

DNA isolation, HPV confirmation, p16, and Ki-67 were evaluated according to the published study [13].

DNA was isolated using the commercially available GeneMATRIX Tissue DNA Purification Kit (Eurx, Gdansk, Poland), after the homogenisation of tumour and margin samples using Lysing Matrix A (MP Biomedicals, Irvine, CA, USA) according to the attached protocol.

The detection of HPV was performed using the GeneFlowTM HPV Array Test Kit (DiagCor Bioscience Ltd., Kowloon Bay, Hong Kong, China), performed by the flow-through system FT-PRO (DiagCor Bioscience Ltd., Kowloon Bay, Hong Kong, China) according to the manufacturer’s instructions.

Ki-67 and 16 were evaluated with immunohistochemistry, using the commercially available kits CONFIRM anti-Ki-67 (30-9) Rabbit Monoclonal Primary Antibody (Ventana Medical Systems, Inc., Tucson, AZ, USA) and CINtec p16 Histology (Roche MTM Laboratories, Mannheim, Germany), respectively.

### 2.3. Statistical Analyses

Shapiro–Wilk was used to test the normality of the results. Student’s *t*-test was used for testing age differences. The U Mann–Whitney test was used for testing differences in protein levels between two groups, and Kruskal–Wallis with Dunn–Sidak post hoc were used to test the significances between more than two groups. The correlations were calculated with Spearman’s rank correlation coefficient. Significant results were considered with *p* < 0.05. Data in the text are presented as median with the 1st and 3rd quartile as follows: median (quartile 1st–quartile 3rd) or mean ± SD. All analyses were conducted with STATISTICA version 13 software (TIBCO Software Inc., Palo Alto, CA, USA).

## 3. Results

### 3.1. Study Group

The study group was composed of 61 HNSCC patients (average age 64 ± 11 years), with 19 (31%) women and 42 (69%) men. Forty-five (73.8%) patients were smokers, and 16 (26.2%) were non-smokers. Overall, 25 patients (41.0%) were occasional drinkers, 13 (21.3%) were regular drinkers, and 23 were abstainers (37.7%). The TNM classification was used to evaluate the tumour samples. The tumour stage was determined in accordance with the 8th edition of the AJCC cancer staging manual [14]. The HNSCC patients included 30 cases of oral squamous cell carcinoma (OSCC), four cases of oropharyngeal squamous cell carcinoma (OPSCC), 20 cases of laryngeal squamous cell carcinoma (LSCC), three cases of hypopharyngeal squamous cell carcinoma (HPSCC), three cases of nasal squamous cell carcinoma (NCSCC), and one case of skin squamous cell carcinoma (SSCC). In the case of cancer classification based on small numbers of some types, only two groups were tested: OSCC (30 cases) and LSCC (20 cases). The clinical characteristics of the study group are presented in Table 1. All patients had no distant metastases (M0).

### 3.2. Protein Level of SPRR and Clinical Parameters

No significant differences were found in the levels of SPRR1A and SPRR2A in HNSCC tumour samples as compared to the margin samples, nor connections with T status. However, there was a lower level of SPRR1A protein in the margin samples of patients with lower nodal status: N0 vs. N1 (0.537 (0.356–0.991) vs. 10.995 (4.060–22.034); *p* = 0.0071). The group with N2 and N3 status were combined because of the small number of N3 cases. The results are presented in Figure 1.

There was a difference in the concentration of the SPRR1A protein in the G2 group between the tumour and margin samples, with a higher level of protein in the tumour (2.092 (1.643–5.316) vs 1.103 (0.370–1.409); *p* = 0.0452). The results are presented in Figure 2. There were no observed differences between the subtypes of HNSCC in tumour and margin samples and also no observed correlation of SPRR proteins with age or sex.

### 3.3. SPRR Protein Level and Tobacco Habit

The median concentration of SPRR1A was significantly higher in non-smoking individuals compared to regular smokers in the margin samples (11.416 (1.360–18.656) vs. 0.656 (3.494–1.237); *p* = 0.0085). Similarly, the median concentration of SPRR2A was higher in non-smokers compared to smokers in the margin samples (13.200 (3.474–23.157) vs. 2.849 (2.356–3.949); *p* = 0.0083). The results are presented in Figure 3 and Figure 4. There was a moderate positive correlation of the SPRR2A protein in the tumour with the number of years a patient smoked (0.3557; *p* = 0.0121).

### 3.4. SPRR Protein Level and Alcohol Habit

The median of the SPRR1A concentration was significantly lower in abstinent individuals, as compared to occasional drinkers, and in regular drinkers compared to occasional drinkers in the margin samples (abstinent vs. occasional: 0.523 (0.343–0.879) vs. 3.501 (1.296–9.322), *p* = 0.0014; regular vs. occasional: 0.304 (0.093–0.944) vs. 3.501 (1.296–9.322), *p* = 0.0135). Furthermore, the median concentration of SPRR2A was significantly lower in abstinent individuals, compared to occasional drinkers, and in regular drinkers compared to occasional drinkers in the tumour samples (abstinent vs. occasional: 1.843 (1.570–3.066) vs 14.376 (2.990–23.801), *p* = 0.0055; regular vs. occasional: 1.905 (0.932–3.253) vs. 14.376 (2.990–23.801), *p* = 0.0113). We observed a significantly lower level of SPRR2A protein in abstinent individuals, as compared to occasional drinkers, and in regular drinkers compared to occasional drinkers in the margins. (abstinent vs. occasional: 2.956 (1.854–3.676) vs. 12.602 (3.970–24.609), *p* = 0.0003; regular vs. occasional: 2.473 (2.375–2.742) vs. 12.602 (3.970–24.609), *p* = 0.0012). The results are presented in Figure 5 and Figure 6. There was also a statistical difference in the age of the abstinent group and the regular drinking group (69 ± 9 vs. 59 ± 10; *p* = 0.0062).

### 3.5. Concentration of SPRR Proteins and HPV Status

There was no difference in SPRR proteins levels according to the p16 status (*p* > 0.05).

The margin samples from HPV-positive patients showed a higher level of SPRR1A, compared to patients with a negative HPV status (2.977 (0.879–6.712) vs. 0.596 (0.349–1.186); *p* = 0.0317) (Figure 7). There was no statistical difference in tumour samples.

### 3.6. SPRR Protein Level in Tumour and Margin Samples in Patients with Concomitant Diseases

We found that the concentration of SPRR1A was significantly higher in tumour samples from patients without concomitant diseases, compared to samples from patients with concomitant diseases (5.893 (1.854–13.560) vs. 0.667 (0.334–1.982); *p* = 0.0165), as presented in Figure 8. A significantly higher concentration of SPRR2A was observed in the margin samples in group without concomitant diseases, compared to patients with concomitant diseases (12.289 (3.449–24.568) vs 2.956 (2.184–4.219); *p* = 0.0145). The results are presented in Figure 9.

### 3.7. SPRR Protein Level in Tumour and Margin Samples According to Ki-67 Proliferation Index Ki-67 Status

No significant differences were found in SPRR1A and SPRR2A levels in HNSCC in connection with the proliferation index Ki-67 < 20 and Ki-67 > 20 status.

## 4. Discussion

Small Proline-Rich Protein 1A and 2A are key proteins associated with the envelope protein of keratinocytes. However, the exact role of SPRR1A and SPRR2A in prognosis and biological function has not been well established in HNSCC. To our knowledge, this has been the first study to analyse the concentrations of these proteins in tumour and surgical margin samples obtained from patients with HNSCC.

Our analysis did not show statistically significant differences in the levels of the SPRR1A and SPRR2A proteins in the tumour samples compared to the margin samples in the HNSCC group of patients. In particular, in our study, in margin samples, significantly higher levels of SPRR1A were found in the group of patients with N1 status than in patients with N0 nodal status. Additionally, we found elevated levels of SPRR1A protein levels in tumour samples, compared to margin samples, in patients with G2 status. Our observations differ from some of the previously published results, in which the authors described elevated or reduced expression of these proteins in tumours compared to non-cancerous samples [12,15,16]. A study by Li et al. compared SPRR1A expression at the mRNA level between tumour samples and healthy control tissue and showed that reduced SPRR1A expression in tumour HNSCC samples was associated with a worse prognosis and lower immune infiltration. In contrast, in another study involving 127 OSCC samples and healthy tissue from adjacent oral areas, the authors found that reduced expression of both SPRR1A and SPRR2A in tumour tissues was associated with the occurrence of lymphatic metastasis [10]. In our study, we used surgical margin tissue, which may have influenced the differences in the results. Margin tissue, although histologically free of tumour cells, may exhibit molecular changes associated with the tumour microenvironment, known as “field cancerization” [17,18]. To date, few studies have focused on SPRR1A expression in the tumour margin in the context of nodal status. However, a study by Braakhuis et al. highlighted that margin tissues may exhibit early molecular changes associated with metastasis, which may indicate active molecular processes that promote disease progression in the margins [18]. In the literature, Michifuri et al. described that SPRR1B (closely related to SPRR1A) is associated with OSCC stem cells and influences tumour cell growth through activation of the MAPK pathway, so it can be suspected that this protein is involved in the regulation of tumour cell differentiation and proliferation [19]. On the other hand, Li et al. showed that in HNSCC, reduced SPRR1A expression correlates with a more aggressive disease course and worse prognosis; so, in our study, elevated SPRR1A in tumours with intermediate differentiation may reflect molecular activity associated with adaptive cellular mechanisms [20].

We found in margin samples that smokers had a significantly lower concentration of SPRR1A and SPRR2A compared to non-smokers in margin samples. Changes in gene and protein expression are crucial in cancer transformation, and the best studied exogenous risk factor is smoking, the components of which include harmful, toxic, carcinogenic, and teratogenic compounds [21]. Components of tobacco smoke can lead to the activation of inflammatory genes, and as a result, numerous pro-inflammatory and anti-inflammatory cytokines are synthesised, promoting the production of SPRRs [6]. On the other hand, a study using rodents exposed to N-nitrosamine, a biomarker of tobacco smoke exposure, found high expression of the *SPRR* gene in the lung [22]. These discrepancies could be due to differences in the tissue type, tested organism, and detection methods. Based on our results, the effect of cigarette smoke components may be through modulation of SPRRs by genetic and epigenetic mechanisms in response to epithelial damage.

In our study, the median concentration of SPRR1A was significantly lower in abstinent individuals, as compared to occasional drinkers, and in regular drinkers compared to occasional drinkers in the margin samples. Furthermore, for SPRR2A, we observed similar correlations in the tumour and margin samples. According to the World Health Organisation (WHO), alcohol ranks third among health risk factors for Europeans. Alcohol toxicity can manifest itself directly, through ethanol, as well as indirectly, through its metabolic products, including acetaldehyde or reactive oxygen species (ROS), resulting in oxidative stress, epigenetic modifications, DNA damage, inaccurate DNA repair, and DNA adduct formation, among others [23]. Occasional alcohol exposure has been shown to have an inhibitory effect on inflammation and immune responses, while regular alcohol exposure has the opposite effect [24]. Furthermore, another study by Demetris et al. suspects that *SPRR2A* gene overexpression may be caused by inflammation and cellular stress involving barrier epithelia [25]. The significant increase in SPRR1A and SPRR2A protein expression in occasional alcohol consumers can be interpreted as an adaptive response to moderate alcohol stress. In regular drinkers, the continuous oxidative stress may lead to the development of other adaptive mechanisms or exhaustion of the ability to increase the expression of these proteins, whereas in abstainers, the lack of stimulus causes expression to remain at baseline levels. We suspect also that there are alternative genetic and epigenetic mechanisms that may be involved in altered gene and protein expression. Further studies are required to more precisely define the mechanisms regulating SPRR1A/SPRR2A expression in the context of different alcohol consumption.

In our study, a significantly higher concentration of SPRR1A was found in margin samples among patients with positive HPV status as compared to margin samples of patients with negative HPV status. The expression of HPV E6 and E7 is closely associated with the epidermis, which is dominated by keratinocytes, a target in the virus growth cycle. Several studies indicate that HPV E6 and/or E7 oncogenes can modulate the expression of genes involved in keratinocyte differentiation [26,27,28,29]. In the case of HPV16-HNSCC patients, a decrease in *SPRR1A* gene expression was noted, according to Pavón et al. [28]. Therefore, according to the concept of “field cancerization” proposed by Slaughter et al. and other studies [30], molecular changes are present in the epithelium surrounding a squamous cell tumour. Therefore, we suggest that there are oncogenic pathways for HPV genes associated with various processes that can modulate the expression of a large number of genes and proteins together with SPRR1A.

Furthermore, we reported the decreased concentration of SPRR1A in tumours and SPRR2A in margin samples in the group of patients with concomitant diseases such as cardiovascular diseases, kidney diseases, gastrointestinal diseases, and endocrine diseases, compared to the group of patients without concomitant diseases. An increase in SPRR protein concentration has been found in a rat in vitro model of ischemic stress, resulting in reduced rates of apoptosis [31]. Similarly, the median concentration of SPRR2A was statistically higher in the serum of subjects with gastritis than in healthy subjects [32]. In another study using a mouse model of chronic kidney disease, a protective function of the SPRR2F protein was observed by preventing oxidative damage [33]. We could search for discrepancies followed by differences in the type of material collected, the type of organism, and finally the detection method. Presumably, changes in the concentration of the SPRR1A and SPRR2A proteins are due to a protective response to keratinocyte damage.

The results of our study indicate that the observed correlations with selected clinical features may have important biological and prognostic significance. From a therapeutic perspective, the identification of patients with elevated SPRR1A levels at N1 and G2 status may help stratify patients requiring a more aggressive therapeutic approach (e.g., more intensive chemoradiotherapy) [10,20]. From a prognostic perspective, the combination of measurements of SPRR1A levels in the margin (as a marker of early “field cancerization” changes) and in the tumour may allow better differentiation of patients in terms of the risk of progression and metastasis [17,18]. The inclusion of such a biomarker in routine testing could support decisions to intensify surgical treatment (e.g., extended lymphadenectomy), plan complementary chemotherapy, or implement immunotherapy in a select group of patients [10,20]. In addition, monitoring SPRR1A levels in the margin could be a useful tool in assessing the risk of micrometastasis or early premalignant lesions, which would allow a more precise mapping of the extent of the tumour lesion when planning surgical circumstances and delineating resection boundaries [17,18,20]. Moreover, in patients with HPV(+), the surgical margin appears to be more molecularly “active”, which may reflect a state of chronic inflammation or repair processes induced by the virus [34]. In a prognostic context, higher SPRR1A levels in the margin from HPV-positive patients may indicate the need for a more thorough evaluation of resection limits and more rigorous monitoring of the patient after surgery, as changes in the margin may indicate subclinical transformations that favour recurrence [20,34].

In our study, we used the surgical margin; however, there were no samples of completely healthy tissue from patients without HNSCC, which is a limitation of our study and may affect the interpretation of the results obtained. Surgical margin tissue, although histologically unaltered, can undergo molecular and genetic modifications due to so-called “field cancerization” [17,18]. It involves a wide area around the tumour showing precancerous or subclinical changes that promote the formation of cancerous foci. Therefore, the expression of the studied SPRR1A and SPRR2A in the margin tissue may differ from their levels in healthy tissue, which may lead to under- or overestimation of the expression differences between the tumour and the margin. Therefore, the results of comparing SPRR1A and SPRR2A levels between tumour and margin samples may not fully reflect the true picture of molecular changes associated with tumour transformation. This phenomenon may explain the lack of statistically significant differences in our analysis. To better define the role and prognostic potential of SPRR1A and SPRR2A proteins, further studies including samples from healthy individuals are needed to establish true reference values and to better interpret whether changes in the expression of these proteins are unique to tumour transformation or reflect processes related to the tumour microenvironment or inflammation.

## 5. Conclusions

We suggest that SPRR1A may be associated with tumour transformation. Moreover, our results suggest that differences in SPRRs are mostly determined by unhealthy behaviours, which, in a wider perspective, could influence carcinogenesis. Small Proline-Rich Protein 1A and 2A levels may be affected by smoking, alcohol consumption, as well as HPV status and concomitant diseases. The small sample size is the main limitation of this study. More studies are needed on a larger group of patients and in more diverse cohorts to verify how changes in SPRR levels can affect tumour development and the prognosis in patients with HNSCC. In further, studies with cell lines and animal models are needed, which could provide valuable information to better understand the role of SPRR1A and SPRR2A proteins in tumorigenesis. Our future studies will then focus on analysing the health and overall survival of patients in the context of the SPRR1A and SPRR2A proteins.

## Figures and Tables

**Figure 1 diagnostics-15-01633-f001:**
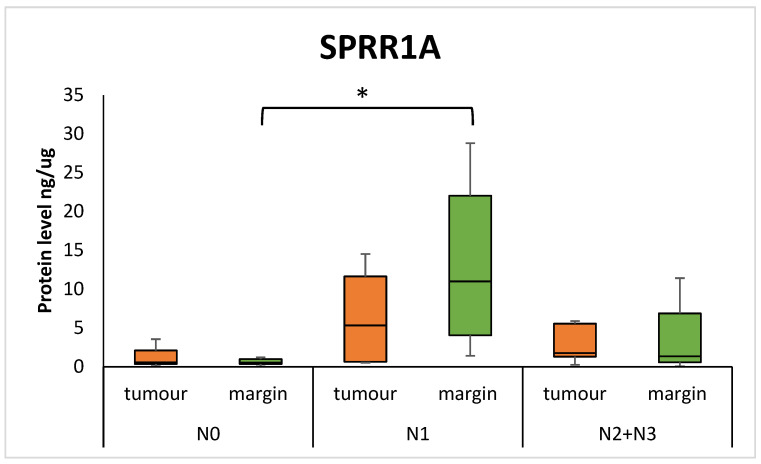
The level of SPRR1A protein in the tumour and margin samples according to the patient’s nodal status. The asterisk indicates a *p*-value < 0.05.

**Figure 2 diagnostics-15-01633-f002:**
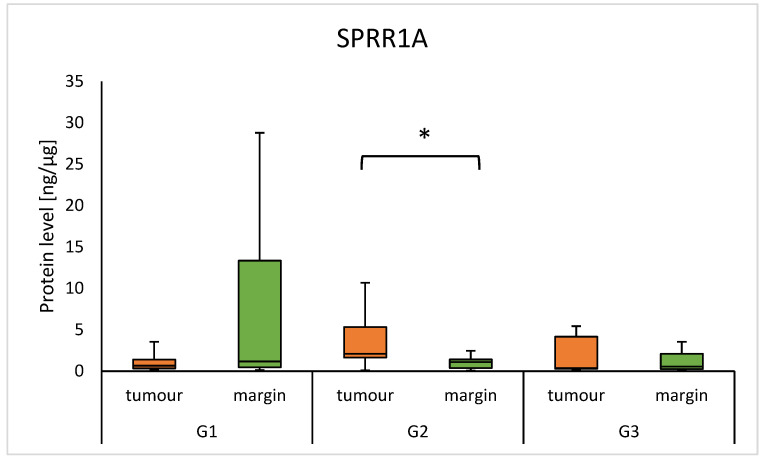
The level of SPRR1A protein in the tumour and margin samples according to grade. The asterisk indicates a *p*-value < 0.05.

**Figure 3 diagnostics-15-01633-f003:**
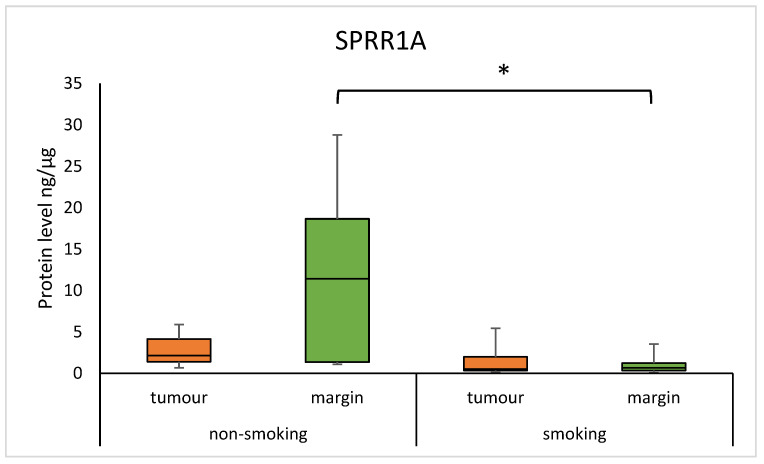
The level of SPRR1A protein in the tumour and margin samples according to smoking status. The asterisk indicates *p*-value < 0.05.

**Figure 4 diagnostics-15-01633-f004:**
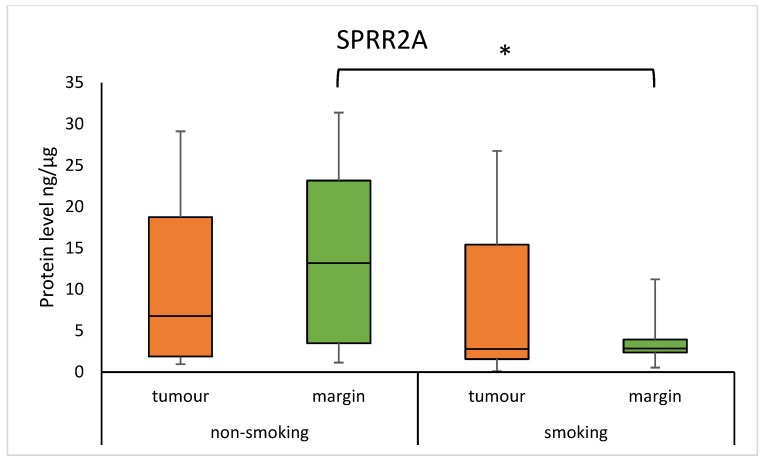
The level of SPRR2A protein in the tumour and margin samples according to smoking status. The asterisk indicates *p*-value < 0.05.

**Figure 5 diagnostics-15-01633-f005:**
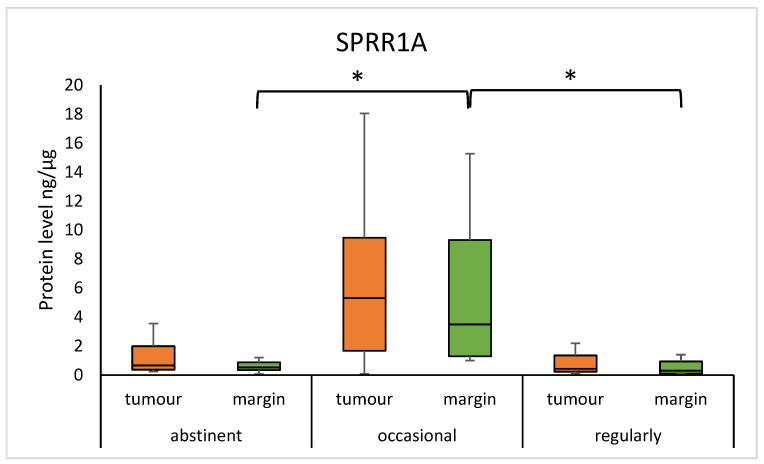
The level of SPRR1A protein in the tumour and margin samples according to drinking status. The asterisks indicate *p*-value < 0.05.

**Figure 6 diagnostics-15-01633-f006:**
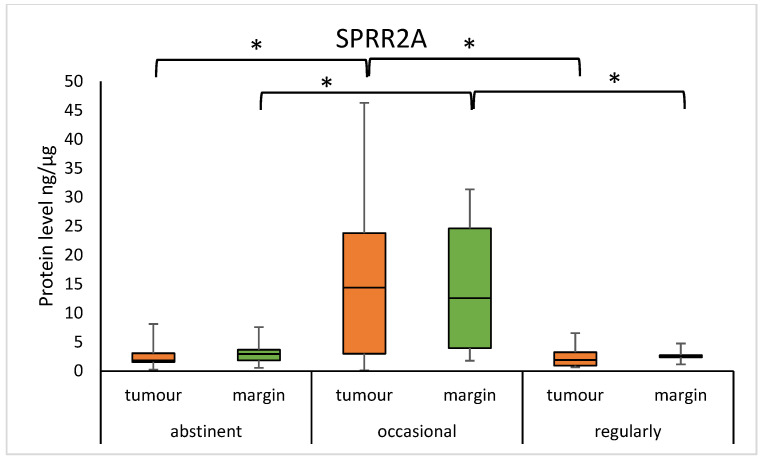
The level of SPRR2A protein in the tumour and margin samples according to drinking status. The asterisks indicate *p*-value < 0.05.

**Figure 7 diagnostics-15-01633-f007:**
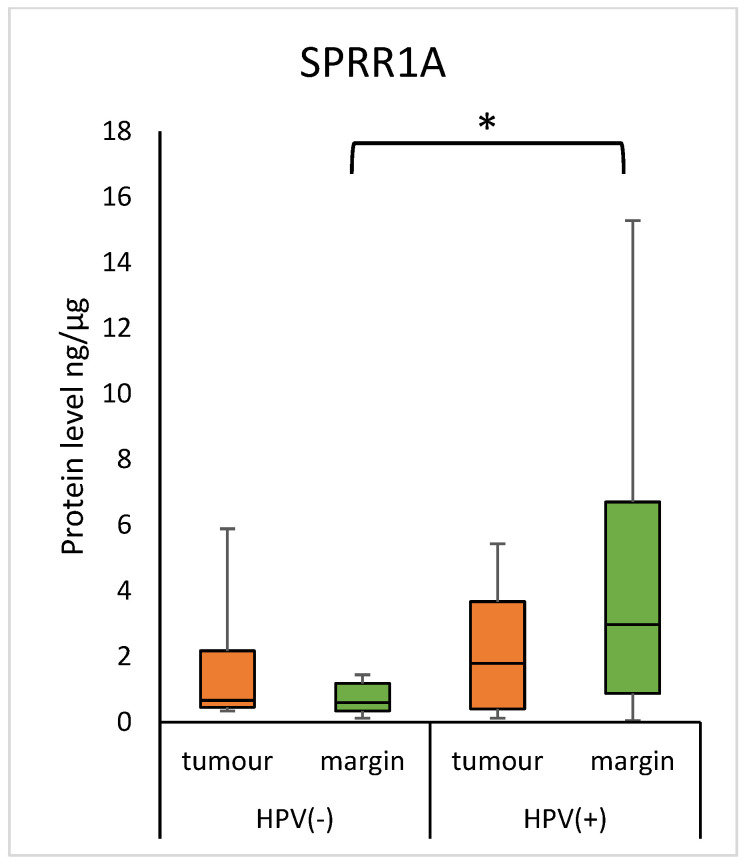
The level of SPRR1A protein in the tumour and margin samples according to HPV status. The asterisk indicates *p*-value < 0.05.

**Figure 8 diagnostics-15-01633-f008:**
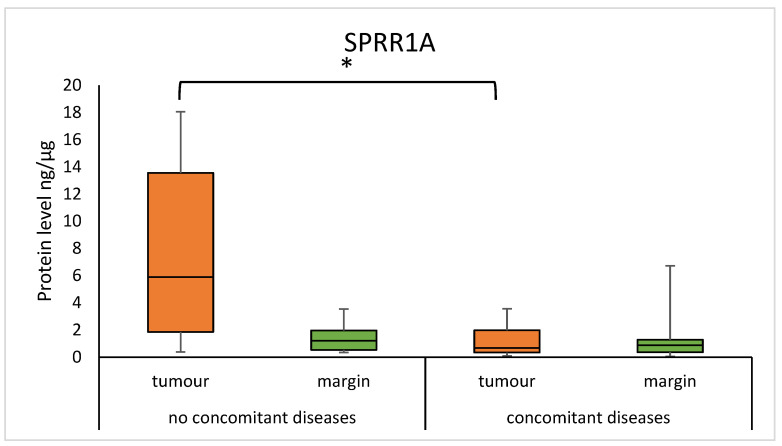
The level of SPRR1A protein in the tumour and margin samples according to the presence of concomitant diseases. The asterisk indicates *p*-value < 0.05.

**Figure 9 diagnostics-15-01633-f009:**
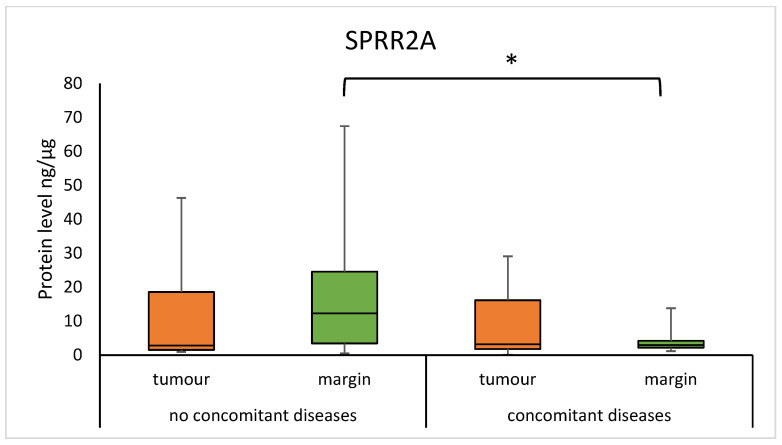
The level of SPRR2A protein in tumour and margin samples according to the presence of concomitant diseases. The asterisk indicates *p*-value < 0.05.

**Table 1 diagnostics-15-01633-t001:** Characteristics of the study group.

Parameter	*n* (%)
T classification	
T1	4 (6.56)
T2	6 (9.84)
T3	21 (34.43)
T4	30 (49.18)
Nodal status (N)	
N0	30 (49.18)
N1	11 (18.03)
N2	17 (27.87)
N3	3 (4.92)
Histological grading (G)	
G1	18 (29.51)
G2	33 (54.10)
G3	5 (8.20)
G4	2 (3.28)
NA *	3 (4.92)
HPV status	
Yes	17 (27.87)
No	29 (47.54)
NA *	15 (24.59)
p16 status	
Yes	12 (19.67)
No	47 (77.05)
NA *	2 (3.28)

* NA—not assessed.

## Data Availability

The data used to support the findings of this research are available upon request.

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
