# Peer review of "Analysis of Selected Small Proline-Rich Proteins in Tissue Homogenates from Samples of Head and Neck Squamous Cell Carcinoma"

_diagnostics, 2025, doi:10.3390/diagnostics15131633_

Round 1
Reviewer 1 Report
Comments and Suggestions for Authors
- The abstract should include what samples were collected, otherwise confusing.
- Were any differences in SPRRs found using the TCGA database? This would be important to include.
- The authors should immunohistochemistry of SPRR1A and SPRR2A in instances where there are significant differences.
- Many of the statistically significant differences appear to be small. Statistical significance does not always mean biological significance.
- In the introduction, the authors reference a paper looking at SPRR levels in oral cavity cancer. This paper should be discussed in comparison to their study in the discussion.
- The authors state "Student's t-test, the U Mann-Whitney test, or Kruskal Wallis... were used..." The paper only provided analysis of ELISA protein levels, so why is more than one test listed. Which was used?
- In table 1, why are there cases where T classification and grading were not assessed? This would be important for patient care. There is also a large number of cases where p16 was not assessed.
- The second paragraph of the discussion does not adequately put their study in the context of previous reports.
There are a few places where there is an extra word or wrong word choice.
Author Response
Thank you kindly for taking the time to review and for your valuable comments, we have sent the answers in the appendix (Word), while the changes to the article are highlighted in yellow.

Reviewer 2 Report
Comments and Suggestions for Authors
This study investigates the concentrations of SPRR1A and SPRR2A in tumour and margin tissues of patients with head and neck squamous cell carcinoma (HNSCC), exploring associations with clinical, demographic, and behavioural variables. The research topic is novel and relevant to understanding the molecular biology of HNSCC. The authors utilize ELISA-based quantification and examine correlations with factors such as smoking, alcohol consumption, HPV, and concomitant diseases.
While the study is commendable in scope and clinical relevance, several methodological, linguistic, and structural aspects need improvement to enhance its scientific impact and clarity.
The introduction mentions a hypothesis regarding altered SPRR levels but lacks a clearly articulated primary objective and hypothesis statement. The authors should define the primary research question more explicitly and state the hypothesis in a dedicated sentence at the end of the Introduction. The grouping of tumour subtypes (e.g., combining LSCC, OPSCC, NCSCC, etc.) is not sufficiently justified. The authors should provide a rationale for this grouping or, if sample sizes allow, analyze them separately to avoid potentially masking subtype-specific trends.
While the statistical tests used are listed, there is no clear explanation for why each test was chosen or whether adjustments for multiple comparisons were applied. Clarify statistical decision-making and include methods such as Bonferroni or FDR correction if multiple hypothesis testing was performed.
Margin tissue is used as a control, but no non-cancer controls are included.This limitation should be discussed more prominently in the Discussion section as it affects the interpretation of baseline protein expression. ç
While several statistically significant differences are reported, the clinical implications of altered SPRR levels are not fully elaborated. The authors should strengthen the discussion around the biological or prognostic significance of SPRR1A/SPRR2A expression patterns in tumour versus margin, particularly regarding their potential role in patient stratification or therapy selection.
Comments on the Quality of English LanguageA thorough English language revision by a native speaker or professional editor is advised. Numerous grammatical errors and awkward phrases reduce readability. Examples include: “is occurs in 95%” → should be “accounts for approximately 95%” or “we were aimed to determine” → “we aimed to determine”
Author Response

(The authors gave the same response as above.)

Round 2
Reviewer 1 Report
Comments and Suggestions for Authors
I feel the authors mainly ignored my suggestions that would increase the strength of the paper. Results from this and previous reports do not seem to have consistency, which limits the clinical utility of SPRRs as prognostic markers.
Reviewer 2 Report
Comments and Suggestions for Authors
The authors have made a thorough and concise revision of the manuscript, following the recommendations of the reviewers, which has succeeded in increasing the quality of the scientific article. In my view, it is ready for publication at the present time.
Author Response
Dear Reviewer,
Thank you for taking the time to review and for all your comments.
Regards,
Authors